# GWAS identifies genetic loci, lifestyle factors and circulating biomarkers that are risk factors for sarcoidosis

Shuai Yuan [1,2,3] ✉, Jie Chen[4], Jiawei Geng [5], Sizheng Steven Zhao[6], James Yarmolinsky[7], Elizabeth V. Arkema [8], Sarah Abramowitz[2], Michael G. Levin [3,9], Kostas K. Tsilidis[7,10], Stephen Burgess [11,12], Scott M. Damrauer [2,3,14] & Susanna C. Larsson [1,13,14]

Sarcoidosis is a complex inflammatory disease with a strong genetic component. Here, we perform a genome-wide association study in 9755 sarcoidosis cases to identify risk loci and map associated genes. We then use transcriptome-wide association studies and enrichment analyses to explore pathways involved in sarcoidosis and use Mendelian randomization to examine associations with modifiable factors and circulating biomarkers. We identify 28 genomic loci associated with sarcoidosis, with the *C1orf141-IL23R* locus showing the largest effect size. We observe gene expression patterns related to sarcoidosis in the spleen, whole blood, and lung, and highlight 75 tissue-specific genes through transcriptome-wide association studies. Furthermore, we use enrichment analysis to establish key roles for T cell activation, leukocyte adhesion, and cytokine production in sarcoidosis. Additionally, we find associations between sarcoidosis and genetically predicted body mass index, interleukin-23 receptor, and eight circulating proteins.

Sarcoidosis is a systemic inflammatory disorder characterized by the emergence of non-caseating granulomas, predominantly in the lungs and lymph nodes[1]. Its prevalence varies globally, ranging from ~1–5 cases per 100,000 individuals in Eastern Asia to 140–160 per 100,000 in Nordic countries[2]. While sarcoidosis can manifest at any age, it predominantly strikes young to middle-aged adults[3,4], profoundly impacting their quality of life, work productivity, and, in later stages,

potentially leading to significant comorbidities and mortality[5–7]. The complex nature of the disease, coupled with its relative rarity and the absence of clear diagnostic measures, obscures its underlying etiology[8], thereby complicating prevention and treatment efforts. Nevertheless, a blend of genetic and environmental factors (e.g., obesity, smoking, and occupational exposures) is proposed to influence development and progression of sarcoidosis[9].

[1]Unit of Cardiovascular and Nutritional Epidemiology, Institute of Environmental Medicine, Karolinska Institutet, Stockholm, Sweden. [2]Department of Surgery, University of Pennsylvania Perelman School of Medicine, Philadelphia, PA, USA. [3]Corporal Michael J. Crescenz VA Medical Center, Philadelphia, PA, USA. [4]Department of Gastroenterology, The Third Xiangya Hospital, Central South University, Changsha, China. [5]Department of Big Data in Health Science School of Public Health, Center of Clinical Big Data and Analytics of The Second Affiliated Hospital, Zhejiang University School of Medicine, Hangzhou, China. [6]Centre for Musculoskeletal Research, Division of Musculoskeletal and Dermatological Science, School of Biological Sciences, Faculty of Biological Medicine and Health, The University of Manchester, Manchester Academic Health Science Centre, Manchester, UK. [7]Department of Epidemiology and Biostatistics, School of Public Health, Imperial College London, London, UK. [8]Department of Medicine Solna, Clinical Epidemiology Division, Karolinska Institutet, Stockholm, Sweden. [9]Division of Cardiovascular Medicine, Department of Medicine, University of Pennsylvania Perelman School of Medicine, Philadelphia, PA, USA. [10]Department of Hygiene and Epidemiology, University of Ioannina School of Medicine, Ioannina, Greece. [11]MRC Biostatistics Unit, University of Cambridge, Cambridge, UK. [12]Department of Public Health and Primary Care, University of Cambridge, Cambridge, UK. [13]Medical Epidemiology, Department of Surgical Sciences, Uppsala University, Uppsala, Sweden. [14]These authors jointly supervised this work: Scott M. Damrauer and Susanna C. Larsson. ✉e-mail: shuai.yuan@ki.se

**Table 1 | Genetic loci associated with sarcoidosis at the genome-wide significance level in the meta-analysis of Europeans and African Americans**

| Lead variant | Position hg38 | Nearby gene | EA | NEA | EAF | OR (95% CI) | P-value | $I^2$(%) | CADD |
|---|---|---|---|---|---|---|---|---|---|
| rs34017352 | 1:67132522 | C1orf141-IL23R | A | C | 0.98 | 0.63 (0.57-0.69) | 3.97E-22 | 52 | 2.08 |
| rs7549723 | 1:150569336 | RN7SL600P | T | C | 0.63 | 1.1 (1.07-1.14) | 3.10E-10 | 0 | 0.44 |
| rs72695390 | 1:167631783 | RCSD1 | T | C | 0.89 | 0.87 (0.83-0.92) | 1.09E-08 | 18 | 5.67 |
| rs2422255 | 1:172815258 | RP1-15D23.2 | T | C | 0.65 | 0.91 (0.88-0.94) | 3.33E-09 | 0 | 6.92 |
| rs34977426 | 2:60731591 | ATP1B3P1 | A | C | 0.17 | 1.12 (1.08-1.17) | 1.54E-09 | 40 | 1.20 |
| rs546039326 | 2:96560781 | ARID5A | A | C | 0.02 | 1.54 (1.35-1.75) | 4.85E-11 | 0 | 11.17 |
| rs145955907 | 2:97725153 | ZAP70 | T | C | 0.02 | 1.56 (1.37-1.77) | 7.23E-12 | 7 | 24.50 |
| rs10183338 | 2:110853239 | ACOXL | T | G | 0.29 | 0.89 (0.86-0.92) | 1.16E-13 | 0 | 5.72 |
| rs67748055 | 2:198046702 | PLCL1 | A | G | 0.44 | 1.1 (1.07-1.13) | 4.87E-11 | 0 | 1.79 |
| rs10189685 | 2:202623726 | AC009960.3 | A | G | 0.29 | 1.13 (1.1-1.17) | 1.34E-15 | 20 | 1.85 |
| rs10933330 | 2:230322452 | SP140 | A | G | 0.74 | 0.91 (0.88-0.94) | 3.89E-08 | 0 | 4.28 |
| rs17041517 | 3:4834831 | ITPR1 | A | G | 0.54 | 0.91 (0.89-0.94) | 4.57E-09 | 16 | 3.49 |
| rs1152002 | 3:12430372 | PPARG | T | C | 0.46 | 1.09 (1.06-1.12) | 1.35E-08 | 0 | 3.94 |
| rs7639471 | 3:112336864 | CD200 | T | G | 0.21 | 0.89 (0.86-0.93) | 9.57E-10 | 0 | 0.89 |
| rs11567997 | 5:138289539 | CDC25C | C | G | 0.91 | 0.84 (0.79-0.9) | 3.12E-08 | 13 | 27.90 |
| rs4921493 | 5:159409099 | AC008703.1 | T | C | 0.68 | 0.92 (0.89-0.95) | 2.38E-08 | 0 | 1.90 |
| rs12195589 | 6:32476807 | HLA-DRB9 | T | C | 0.41 | 0.7 (0.67-0.72) | 9.71E-114 | 90 | 3.09 |
| rs873973 | 7:75837299 | CCL24 | T | C | 0.18 | 1.14 (1.1-1.18) | 2.57E-12 | 32 | 2.36 |
| rs77112238 | 9:126425324 | MVB12B | A | G | 0.90 | 0.88 (0.84-0.92) | 4.43E-08 | 8 | 1.80 |
| rs77029323 | 10:62690119 | ZNF365 | T | G | 0.35 | 1.11 (1.07-1.14) | 6.48E-10 | 13 | 18.03 |
| rs11202051 | 10:80193615 | ANXA11 | A | G | 0.41 | 0.78 (0.76-0.8) | 6.41E-58 | 67 | 0.21 |
| rs663743 | 11:64340263 | CCDC88B | A | G | 0.34 | 0.85 (0.82-0.88) | 3.68E-22 | 0 | 5.93 |
| rs7965287 | 12:57794913 | TSFM | C | G | 0.33 | 0.91 (0.88-0.94) | 4.19E-10 | 0 | 0.64 |
| rs3184504 | 12:111446804 | SH2B3 | T | C | 0.41 | 1.17 (1.14-1.21) | 1.14E-23 | 11 | 11.21 |
| rs11856316 | 15:82846540 | HOMER2 | A | C | 0.65 | 1.1 (1.07-1.13) | 1.08E-09 | 0 | 2.07 |
| rs4788115 | 16:28986790 | LAT | A | T | 0.22 | 1.13 (1.09-1.18) | 8.13E-11 | 0 | 8.25 |
| rs34536443 | 19:10352442 | TYK2 | C | G | 0.04 | 0.68 (0.62-0.74) | 4.55E-16 | 0 | 25.50 |
| rs1893592 | 21:42434957 | UBASH3A | A | C | 0.71 | 1.1 (1.06-1.14) | 2.78E-08 | 0 | 11.16 |

Statistical analyses for GWAS associations were conducted using a two-sided test, with genome-wide significance thresholds ($P < 5 \times 10^{-8}$) applied to account for multiple comparisons. *CADD* Combined Annotation Dependent Depletion, *CI* confidence interval, *EA* effect allele, *EAF* effect allele frequency, *NEA* non-effect allele, *OR* odds ratio.

In recent decades, considerable efforts have been directed toward unveiling the genetic foundation of sarcoidosis[10]. Several genetic loci nearby genes like *MHC*[11–13], *ANXA11*[13,14], *CCDC88B*[15], and *IL23*[16,17], have been linked to sarcoidosis susceptibility. In addition, genome-wide association study (GWAS) and expression quantitative trait loci analyses have identified more associated genes, like *ADCY3* and *CCL24*[12,18]. However, many of these earlier investigations might have been limited by modest sample sizes[11] or the dependence on targeted genotyping platforms such as ImmunoChip[12,16,19], potentially overlooking broader genetic variations associated with sarcoidosis risk.

Mendelian randomization (MR) is an epidemiological approach that can strengthen causal inference by leveraging genetic variants as the instrumental variable for the exposure[20]. It has been successfully employed across a range of diseases to establish casual risk factors and to identify targets for therapeutic manipulation[21,22]. Due to the relatively modest size and power of previous sarcoidosis GWAS, it has not been possible to apply MR approaches for causal inference with respect to sarcoidosis.

To address these gaps, we conducted a GWAS meta-analysis, harnessing data from four large-scale studies including data from Europeans and African Americans, aiming to capture a more comprehensive genetic landscape of sarcoidosis. Moreover, we performed transcriptome-wide association study (TWAS) to identify associated genes and enriched molecular pathways based on these signals. We further conducted MR analyses to uncover the potential causal roles of important modifiable factors (adiposity measures and lifestyle factors), inflammatory markers, and circulating proteins in sarcoidosis with the aim of highlighting potential preventive and therapeutic targets.

## Results

### Genome-wide association analysis

This GWAS meta-analysis included 9755 individuals with sarcoidosis (7554 cases of European and 2201 cases of African Americans) and 1,526,867 individuals without sarcoidosis. Assuming a lifetime prevalence of 0.1%[23], the heritability of sarcoidosis on liability scale was estimated at $h^2 = 9.3\%$ (95% CI 7.1%–11.6%). The genomic inflation factor, $\lambda_{GC}$, was 1.11, suggesting a minimal inflation due to population stratification or other confounding factors, which was also observed in the quantile-quantile plot (Supplementary Fig. 1). The LD (linkage disequilibrium) Score regression (LDSC) intercept was 1.03 (95% CI 1.01-1.05) and the LDSC ratio was 0.17 (95% CI 0.07-0.27), indicating the majority of the signal is not likely to driven by population-related biases.

### Loci identification and gene annotation

Functional Mapping and Annotation (FUMA) analysis annotated 28, 24, and 3 genomic loci that surpassed the genome-wide significance threshold ($P < 5 \times 10^{-8}$) in GWAS including all participants (Table 1), only those with European ancestry (Table 2), and only those with African Americans (Table 3) (Supplementary Fig. 2). The regional plots for all these loci are presented in Supplementary Figs. 3–6. The locus

**Table 2 | Genetic loci associated with sarcoidosis at the genome-wide significance level among Europeans**

| Lead variant | Position hg38 | Nearby gene | EA | NEA | EAF | OR (95% CI) | P-value | I²(%) | CADD |
|---|---|---|---|---|---|---|---|---|---|
| rs2024825 | 1:67132294 | C1orf141-IL23R | T | C | 0.17 | 1.26 (1.21-1.31) | 8.79E-27 | 0 | 3.43 |
| rs10888386 | 1:150639481 | GOLPH3L | A | G | 0.35 | 0.9 (0.87-0.94) | 6.86E-09 | 0 | 5.66 |
| rs72695390 | 1:167631783 | RCSD1 | T | C | 0.88 | 0.87 (0.83-0.91) | 4.04E-08 | 45 | 5.67 |
| rs1102730 | 1:172802420 | RP1-15D23.2 | T | C | 0.62 | 0.91 (0.88-0.94) | 4.96E-08 | 0 | 1.47 |
| rs11694327 | 2:26696561 | KCNK3 | T | C | 0.74 | 0.9 (0.87-0.93) | 2.39E-08 | 71 | 13.39 |
| rs6761132 | 2:60754753 | PAPOLG | T | C | 0.17 | 1.14 (1.09-1.19) | 5.46E-10 | 47 | 0.58 |
| rs546039326 | 2:96560781 | ARID5A | A | C | 0.02 | 1.54 (1.35-1.75) | 4.85E-11 | 0 | 11.17 |
| rs145955907 | 2:97725153 | ZAP70 | T | C | 0.02 | 1.56 (1.37-1.77) | 7.23E-12 | 7 | 24.50 |
| rs10183338 | 2:110853239 | ACOXL | T | G | 0.29 | 0.88 (0.85-0.91) | 1.90E-12 | 0 | 5.72 |
| rs1401090 | 2:197943854 | PLCL1 | A | G | 0.43 | 1.11 (1.08-1.15) | 9.59E-11 | 0 | 0.62 |
| rs191390916 | 2:202715143 | FAM117B | T | C | 0.29 | 1.15 (1.11-1.19) | 5.39E-15 | 0 | 1.08 |
| rs17041517 | 3:4834831 | ITPR1 | A | G | 0.56 | 0.91 (0.88-0.94) | 1.37E-08 | 0 | 3.49 |
| rs12195589 | 6:32476807 | HLA-DRB9 | T | C | 0.44 | 0.67 (0.65-0.69) | 4.18E-116 | 75 | 3.09 |
| rs7811626 | 7:75823936 | CCL24 | T | G | 0.19 | 1.15 (1.11-1.2) | 3.87E-12 | 15 | 1.55 |
| rs57057378 | 10:13118035 | OPTN | A | G | 0.24 | 1.12 (1.08-1.17) | 2.30E-09 | 54 | 12.51 |
| rs77029323 | 10:62690119 | ZNF365 | T | G | 0.37 | 1.1 (1.07-1.14) | 4.76E-09 | 44 | 18.03 |
| rs1049550 | 10:80166946 | ANXA11 | A | G | 0.44 | 0.78 (0.76-0.81) | 3.15E-49 | 66 | 28.60 |
| rs663743 | 11:64340263 | CCDC88B | A | G | 0.36 | 0.85 (0.82-0.88) | 3.90E-21 | 0 | 5.93 |
| rs4766578 | 12:111466567 | ATXN2 | A | T | 0.54 | 0.85 (0.83-0.88) | 3.30E-22 | 0 | 1.68 |
| rs11856316 | 15:82846540 | HOMER2 | A | C | 0.63 | 1.1 (1.06-1.14) | 3.02E-08 | 0 | 2.07 |
| rs4788115 | 16:28986790 | LAT | A | T | 0.23 | 1.13 (1.08-1.17) | 1.24E-09 | 0 | 8.25 |
| rs34536443 | 19:10352442 | TYK2 | C | G | 0.04 | 0.67 (0.61-0.74) | 1.72E-15 | 0 | 25.50 |
| rs1893592 | 21:42434957 | UBASH3A | A | C | 0.69 | 1.11 (1.07-1.15) | 1.16E-08 | 0 | 11.16 |
| rs757870 | 22:30380430 | RNF215 | T | C | 0.63 | 0.9 (0.87-0.93) | 3.29E-09 | 57 | 0.30 |

Statistical analyses for GWAS associations were conducted using a two-sided test, with genome-wide significance thresholds ($P < 5 \times 10^{-8}$) applied to account for multiple comparisons. *CADD* Combined Annotation Dependent Depletion, *CI* confidence interval, *EA* effect allele, *EAF* effect allele frequency; *NEA* non-effect allele, *OR* odds ratio.

**Table 3 | Genetic loci associated with sarcoidosis at the genome-wide significance level among African Americans**

| Lead variant | Position hg38 | Nearby gene | EA | NEA | EAF | OR (95% CI) | P-value | I²(%) | CADD |
|---|---|---|---|---|---|---|---|---|---|
| rs11465820 | 1:67255707 | C1orf141-IL23R | A | G | 0.02 | 2.08 (1.66-2.6) | 2.03E-10 | 0 | 1.00 |
| rs1140310 | 6:32665006 | HLA-DQB1 | A | C | 0.73 | 0.67 (0.62-0.72) | 1.44E-27 | 18 | 8.26 |
| rs2573348 | 10:80166029 | ANXA11 | T | C | 0.20 | 0.73 (0.66-0.8) | 2.77E-11 | 0 | 4.27 |

Statistical analyses for GWAS associations were conducted using a two-sided test, with genome-wide significance thresholds ($P < 5 \times 10^{-8}$) applied to account for multiple comparisons. *CADD* Combined Annotation Dependent Depletion, *CI* confidence interval, *EA* effect allele, *EAF* effect allele frequency, *NEA* non-effect allele, *OR* odds ratio.

near the *C1orf141-IL23R* gene had the largest effect size (Supplementary Fig. 7). Per additional minor allele (C allele), the odds ratio (OR) of sarcoidosis was 1.59 (95% CI 1.45-1.75). The lead variant proximal to *CDC25C* and *TYK2* had a Combined Annotation Dependent Depletion (CADD) score more than 25.5, highlighting its potential deleterious impact on protein function and underscoring its importance in genetic susceptibility (Table 1). Associations with identified 28 sarcoidosis-associated loci in the GWAS Catalog database are shown in Supplementary Data 1. Many of these loci were associated with other auto-immune diseases, such as Crohn's disease, rheumatoid arthritis, multiple sclerosis, and celiac disease (Supplementary Data 2) as well as with immune cells, such as lymphocyte and eosinophil counts.

## TWAS and pathway enrichment

MAGMA (Multi-marker Analysis of GenoMic Annotation) tissue expression analysis revealed notable gene expression patterns in spleen, whole blood, and lung ($P_{bon} < 0.05$) (Supplementary Fig. 8). Based on GTEx v8 gene expression data of these tissues, TWAS identified that genetically predicted expression of 60, 45, and 50 genes in spleen, whole blood, and lung respectively were associated with the risk of sarcoidosis after Bonferroni multiple testing correction

(Supplementary Data 3). Figure 1a–c shows the top 20 gene-sarcoidosis associations from TWAS in spleen, whole blood, and lung. Among them, there were 14 genes (*ABT1, BTN2A2, C2orf74, CCDC88B, FAM213A, MDH2, METTL21B, PRSS16, RP3-473L9.4, TCF19, TSFM, UBASH3A, UQCC2, ZSCAN26*) associated with sarcoidosis risk consistently across these tissues. Based on a total of 97 unique TWAS signals, the GO (Gene Ontology) database enrichment analysis identified several relevant pathways (Fig. 1d) potentially associated with sarcoidosis development including regulation of T cell and lymphocyte activation, leukocyte cell-cell adhesion, and activities related to MHC protein complex (Fig. 1e).

## Mendelian randomization analysis

Of the 6 evaluated modifiable risk factors, genetically predicted BMI, WHR and MVPA showed associations with sarcoidosis in the primary analysis (Fig. 2). A per standard deviation (SD) increase in genetically predicted body mass index (BMI) and waist-to-hip ratio (WHR) was associated with a 1.26-fold (95% CI, 1.10-1.45; $P = 0.001$) and 1.17-fold (95% CI 1.03-1.33; $P = 0.016$) increase in the odds of sarcoidosis. In contrast, having at least moderate-to-vigorous physical activity (MVPA) was associated with a 34% decreased risk (OR = 0.66; 95% CI

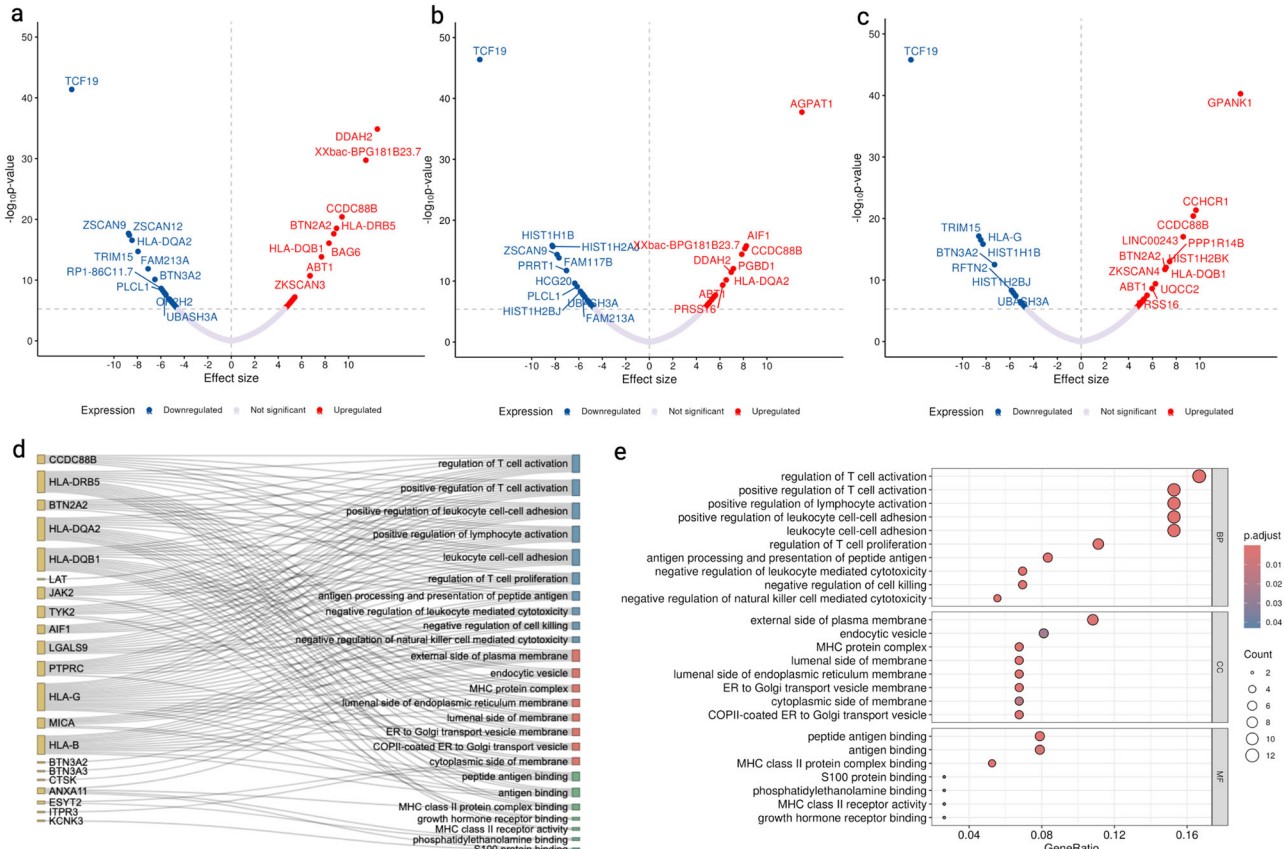

Fig. 1 | Transcriptome-wide association study (TWAS) of sarcoidosis and pathway enrichment. a TWAS in spleen. b TWAS in whole blood. c TWAS in lung. In a–c, statistical analyses were conducted using a two-sided test, with Bonferroni correction applied for multiple comparisons, and the top 20 signals were labeled. d genes identified by TWAS associated with pathways. e GO pathway enrichment based on TWAS signals survived after Bonferroni correction. GO enrichment analysis was performed using a two-sided test, with false discovery rate (FDR) correction applied to account for multiple comparisons. Source data are provided as a Source Data file.

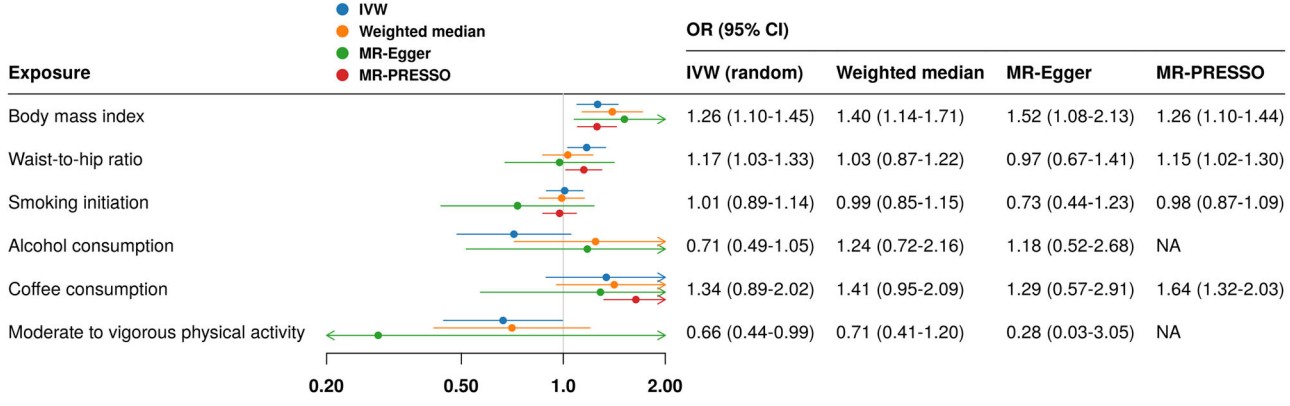

Fig. 2 | Associations between genetically predicted lifestyle factors and risk of sarcoidosis in Mendelian randomization (MR) analysis. Statistical analyses were conducted using a two-sided test. Data are presented as odds ratio +/− 95% confidence interval. The analysis was based on European GWAS data including 7554 cases 1,367,006 and controls. The number of genetic instruments for these traits is presented in Supplementary Data 7. Source data are provided as a Source Data file. CI, confidence interval; IVW, inverse variance weighted with multiplicative random effects; OR, odds ratio.

0.44-0.99; *P* = 0.047) compared with inactivity. After false discovery rate (FDR) correction, only the BMI-sarcoidosis association retained its significance. These associations remained directionally consistent in the sensitivity analyses (Fig. 2). While there was moderate-to-high heterogeneity in these associations except for MVPA, no evident horizontal pleiotropy was identified by MR-Egger intercept test (*P* > 0.05). Genetically predicted levels of other modifiable risk factors showed no significant associations with sarcoidosis risk (Fig. 2).

One inflammatory marker was excluded due to the absence of SNPs in the outcome GWAS and the unavailability of a suitable proxy, leaving 65 markers in the analysis. Genetically predicted levels of 10 inflammatory markers were associated with the risk of sarcoidosis (Fig. 3a, b). Upon adjustment for multiple testing, the associations for interleukin-23 receptor, serum amyloid A (SAA)-2 protein, interleukin-12 receptor subunit beta-2, and interleukin-2 receptor subunit beta persisted. For each SD increase in genetically predicted levels of these

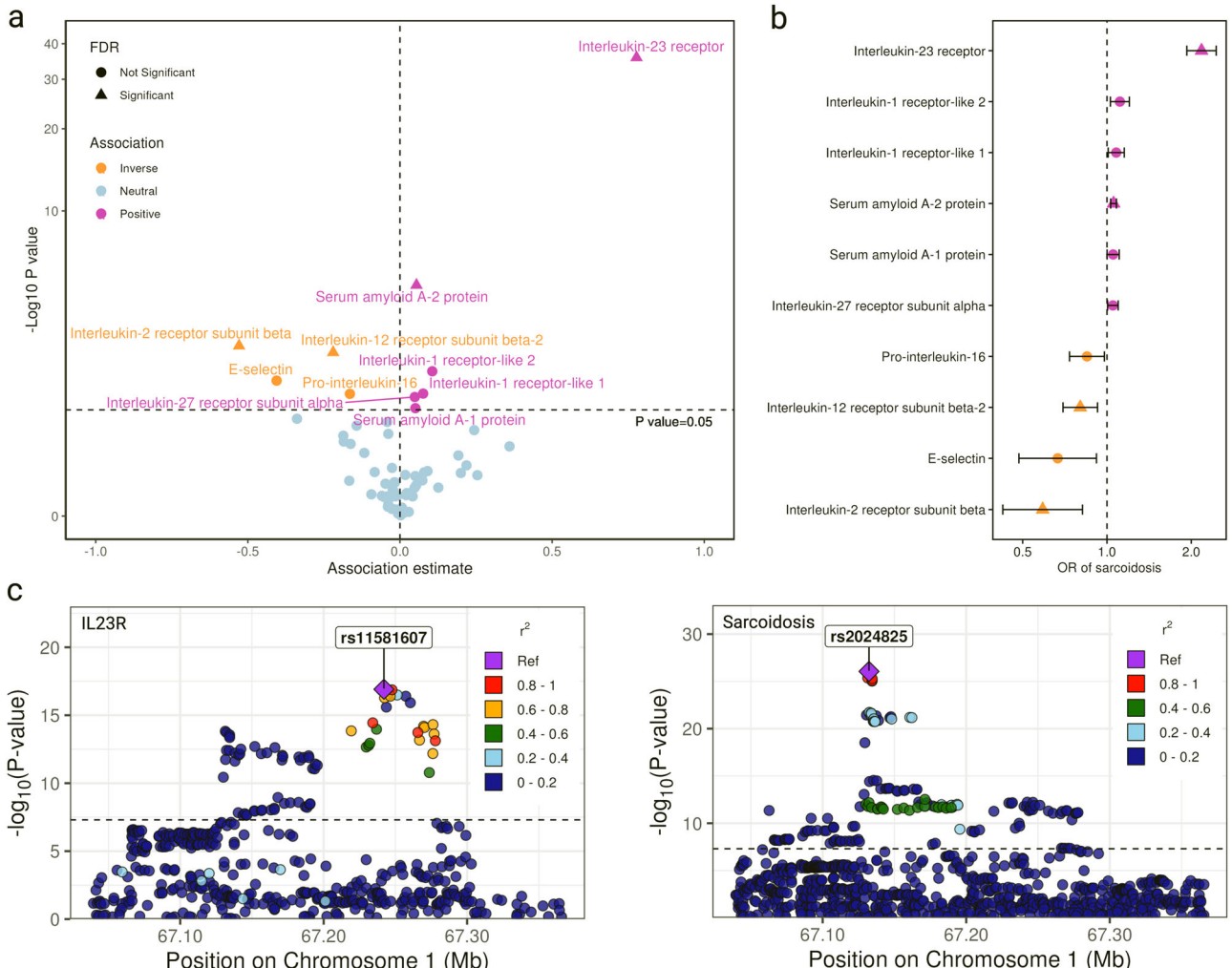

**Fig. 3 | Associations between genetically predicted levels of 65 inflammatory markers and the risk of sarcoidosis in Mendelian randomization (MR) analysis.** Statistical analyses were conducted using a two-sided test. The analysis was based on European GWAS data including 7554 cases 1,367,006 and controls. The number of genetic instruments for inflammatory markers is presented in Supplementary Data 4. **a** Volcano plot of MR associations between 65 inflammatory markers and sarcoidosis. **b** Forest plot of 10 associations with *P*-value < 0.05. Data are presented as odds ratio +/− 95% confidence interval. **c** Reginal locus plot for interleukin 23 receptor (IL23R) and sarcoidosis. The associations in a and b were scaled to one standard deviation increase in genetically predicted markers. Source data are provided as a Source Data file. OR, odds ratio.

proteins, the OR of sarcoidosis was 2.18 (95% CI 1.93-2.46; $P = 1.69 \times 10^{-36}$), 1.06 (95% CI 1.03-1.08; $P = 7.31 \times 10^{-6}$), 0.80 (95% CI 0.70-1.93; $P = 2.43 \times 10^{-3}$), and 0.59 (95% CI 0.42-0.82; $P = 1.56 \times 10^{-3}$), respectively. Comprehensive results for all 65 inflammatory markers can be found in Supplementary Data 4. Even though moderate colocalization evidence was detected for interleukin-23 receptor using the traditional colocalization method (posterior probabilities H4 = 39%), the locus plot identified two signal peaks for interleukin-23 receptor (Fig. 3c). Using the Sum of Single Effects (SuSiE) method, strong colocalization evidence was observed for interleukin-23 receptor at the one peak (rs2024825, posterior probabilities H4 = 99%) but not at the other (rs11581607, posterior probabilities H4 = 0%). Moderate colocalization evidence was detected for interleukin-2 receptor subunit beta (posterior probabilities H4 = 66%) and weak colocalization evidence was detected for the other associations (Supplementary Data 5).

Protein-wide MR analysis identified associations of genetically predicted levels of 19 proteins with sarcoidosis risk after multiple testing correction (Fig. 4a). The OR of sarcoidosis ranged from 0.02 (95% CI 0.00, 0.16) to 6.89 (95% CI 5.23,9.08) per SD increase in genetically predicted levels of protein CSNK2B and AIF1, respectively

(Fig. 4b). Nine proteins with available genetic instruments from the Fenland study and seven protein-sarcoidosis associations were replicated using this protein data source (Supplementary Data 6). Among 19 protein-sarcoidosis associations, strong genetic colocalization evidence was observed for 8 pairs, including BTN3A3, ANXA11, ITPKA, BTN3A1, G3BP1, IL1RN, IL2RB, and NFKB1 (Fig. 4c).

## Discussion
The GWAS of 9755 sarcoidosis cases identified 28 susceptibility loci, revealing key insights into the genetic basis of sarcoidosis. Gene prioritization revealed the largest effect size of the lead variant near *C1orf141-LR23R* gene and the high potential deleterious impact of the lead variants near *CCDC88* and *TYK2* genes, suggesting a substantial impact of these genes on disease susceptibility. Tissue-specific analyses highlighted the importance of sarcoidosis-associated gene expression in the spleen, whole blood, and lung, underscoring their potential roles in sarcoidosis pathogenesis. Additionally, MR analysis suggested potential causal associations of BMI, interleukin-23 receptor, and multiple circulating proteins with sarcoidosis risk, pointing to critical pathways for further exploration.

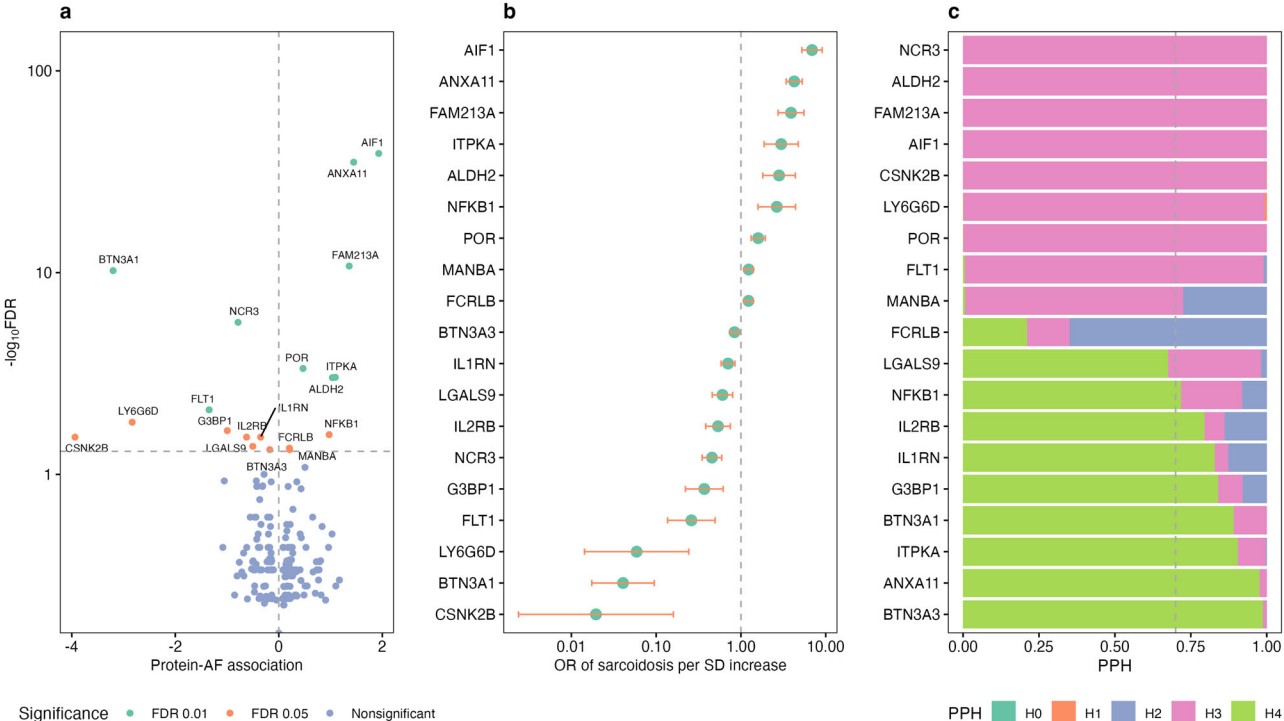

**Fig. 4 | Protein-wide Mendelian randomization analysis of sarcoidosis and corresponding colocalization analysis. a** Volcano plot of genetically predicted levels of proteins in relation to sarcoidosis. The associations with false discovery rate (FDR) < 0.05 were labeled. **b** Forest plot of the associations survived after multiple testing correction. Data are presented as odds ratio +/− 95% confidence interval. The analysis was based on European GWAS data including 7554 cases 1,367,006 and controls. The levels of proteins were genetically instrumented by one lead cis genetic variant. **c** Colocalization analysis of identified protein-AF association. Posterior probabilities (PP) H0 stands for no association between the genetic variants and either trait. PPH1 stands for an association between the genetic variants and only the protein, with no association with sarcoidosis. PPH2 stands for an association between the genetic variants and only sarcoidosis, with no association with the protein. PPH3 stands for an association between the genetic variants and both traits, but with independent signals. PPH4 stands for colocalization, where a shared genetic signal is associated with both traits. PPH4 > 0.7 was deemed strong colocalization evidence. Source data are provided as a Source Data file.

This study confirmed 11 previously reported sarcoidosis-associated loci near *IL23R*[16,17], *FAM117B*[16], *SP140*[16], *PPARG*[24], *IL12B*[16], *HLA-DQA1*[11–13], *CCL24*[17], *ANXA11*[12,13,17,25], *CCDC88B*[16], *SH2B3*[16], and *TYK2*[16] in independent data. By including more sarcoidosis cases, the study identified 17 unreported loci associated with sarcoidosis. Even though the mechanisms underlying the associations are unknown, many of these loci have been associated with white blood cell traits and other autoimmune diseases, revealing the central role inflammation and immune related pathways. In addition, our TWAS pinpointed many tissue-specific sarcoidosis-associated genes and found expressions of 14 genes (*ABT1, BTN2A2, C2orf74, CCDC88B, FAM213A, MDH2, METTL21B, PRSS16, RP3-473L9.4, TCF19, TSFM, UBASH3A, UQCC2, ZSCAN26*) associated with sarcoidosis across all three target tissues. Among these, both GWAS and TWAS found a potential role of *CCDC88B* in sarcoidosis. The mechanisms explaining this association may be related to T-cell maturation, cytokine production, and cell migration, thus leading to granuloma formation[26]. However, this gene has been associated with several other autoimmune diseases, including psoriasis, primary biliary cirrhosis, multiple sclerosis, and inflammatory bowel disease[27], indicating its universal role in overall autoimmune disorders.

GWAS identified a lead variant near *C1orf141-IL23R* associated with sarcoidosis with the largest effect size, particularly in people with African Americans. This gene has also been linked with sarcoidosis Eastern Asians as well[17]. Functional predictions and protein network analyses underscore the significance of the drug targetable IL23/Th17 signaling pathway in the genetic underpinnings of sarcoidosis[16]. Furthermore, our MR analysis revealed that genetically predicted higher levels of the interleukin-23 receptor were associated with a heightened risk of sarcoidosis. Ustekinumab, a monoclonal antibody that targets the p40 subunit of IL-12/IL-23, showed no efficacy in treating pulmonary sarcoidosis. However, this may not reflect direct targeting of the IL23 receptor and null results should be interpreted in the context of challenges in trial design (e.g., choice of surrogate outcomes and potential masking effects of concurrent glucocorticoids).

*TYK2*, encoding a member of the Janus kinase (JAK) family involved in IL23 signaling, has been recognized as an important genetic locus in various autoimmune diseases, including sarcoidosis[16]. Selective inhibitors of TYK2, like Deucravacitinib, have gained approval for treating moderate-to-severe plaque psoriasis and are considered promising targets for multiple autoimmune conditions[28]. An MR study showed that genetically proxied TYK2 inhibition was associated with reduced sarcoidosis susceptibility[29]. A preliminary trial demonstrated that Tofacitinib, a non-selective JAK inhibitor that has some action on TYK2, led to significant improvement in patients with cutaneous sarcoidosis[30]. However, unlike the "too strong" immune response in autoimmune disease, inherited *TYK2* deficiency impairs IL-12/-23-dependent IFN-γ immunity and is associated with an elevated risk of Mycobacterium tuberculosis infection[31]. This gene defect was observed in 1% of tuberculosis cases[31] and probably more patients with nontuberculous mycobacterial infection[32]. Thus, clinical trials to evaluate the safety and efficacy of TYK2 inhibitors in treating sarcoidosis are urgently needed for therapeutic development. In addition, the association between *TYK2* and autoimmune diseases appear to be ancestry-specific[33].

Pathway enrichment based on TWAS signals suggested that T cell and lymphocyte activation, and leukocyte cell-cell adhesion played roles in sarcoidosis. These findings not only uncover the etiology of the disease but provide evidence for disease treatment and management. For example, commonly used as the first-line treatment for

sarcoidosis, corticosteroids reduce inflammation[34]. It also hints the potentials of other treatments for sarcoidosis, such as infliximab and adalimumab[34].

Existing evidence regarding the link between obesity and sarcoidosis is limited and inconclusive[23,35–37]. While two cohort studies involving a large number of women identified a positive relationship between weight gain and adult obesity with heightened sarcoidosis incidence[35,36], a nested case-control study observed a positive association for obesity but not for overweight[38] and a retrospective case-control study observed no significant association between elevated BMI and sarcoidosis risk[37]. Using the MR approach, our research identified significant associations between genetically predicted BMI and sarcoidosis. A broader association between adiposity and sarcoidosis is further supported by suggestive links between genetically predicted waist-to-hip ratio (WHR) and sarcoidosis, although the effect size was smaller compared to that of BMI. This suggests that BMI may serve as a more robust proxy for obesity in distinguishing sarcoidosis risk at the population level. Owing to its proficiency in mitigating confounding and reverse causation, our MR results reinforce the notion that excessive body weight may escalate sarcoidosis risk.

The relationship between serum biomarkers and sarcoidosis has been explored in many studies even though most studies were based on prevalent sarcoidosis[39]. A comprehensive review on biomarkers for the diagnosis and prognosis of sarcoidosis identified soluble interleukin-2 receptor and SAA as the most sensitive markers for confirming the disease[39]. While our study focused on causal inference, our findings also support the involvement of the interleukin-2 pathway and serum amyloid A in sarcoidosis. Furthermore, our protein-wide MR analysis identified some other proteins potentially causally associated with sarcoidosis. For example, NFKB1 plays an important role in NF-κB signaling and corresponding gene has been associated with sarcoidosis[16]. Targeting NF-κB pathways is a well-established therapeutic approach in inflammatory diseases. Drugs that inhibit NF-κB signaling, such as corticosteroids and other immunomodulators, are already used in sarcoidosis treatment. IL1RN has been implicated in sarcoidosis through its role in modulating inflammatory responses by antagonizing IL-1[40]. Anakinra, an IL-1 receptor antagonist, is already used in treating various inflammatory conditions and might be repurposed for sarcoidosis treatment[41]. Both GWAS and protein-wide MR found the signal of ANXA11. The mutation of this gene has been associated with dysregulation of calcium homeostasis and stress granule dynamics[42] that may contribute to the chronic inflammation and granuloma formation. However, no known drug targets exist for this protein.

The strengths of this study include a comparatively large number of sarcoidosis cases, a standardized approach to loci annotation and gene mapping, and a comprehensive investigation of modifiable risk factors in relation to sarcoidosis risk. We also presented data for both Europeans and African Americans, which deepens the understanding of genetic structures of sarcoidosis across populations.

There are limitations of the study. First, combining data from the different populations might introduce population stratification bias due to different genetic background. However, population principal components were taken into consideration before GWAS meta-analysis. In addition, this approach has been often used in previous studies[43,44] and limited heterogeneity was observed between studies. Second, while our GWAS included both European and African American populations, our TWAS and MR analyses primarily focused on individuals of European ancestry. This decision was made to minimize population structure bias and address limitations in statistical power. Future studies are needed to extend these analyses to other populations for a more comprehensive understanding. Third, some misclassification of the outcome was inevitable as sarcoidosis is difficult to diagnose and the diagnostic codes cannot be used to identify sarcoidosis phenotypes. Fourth, our study cannot differentiate genetic associations for sarcoidosis with varying clinical features (e.g., sarcoidosis with and without Löfgren's syndrome) due to data unavailability. Although we fully support further GWAS on sarcoidosis subtypes, we believe our paper contributes meaningfully to understanding both genetic and non-genetic risk factors for sarcoidosis. Fifth, our analytic power might be inadequate particularly for loci or MR associations with weak to moderate effect sizes although this is hitherto the largest GWAS of sarcoidosis. Sixth, this study was based on a series of in silico analysis. Whether the identified loci have causal roles in sarcoidosis needs multidimensional data, like experimental studies. Seventh, we were unable to delve into sex-specific associations due to data constraints and potential power challenges. Last, the fixed number of genetic principal components in the included studies could not be adjusted due to the unavailability of individual-level data. This constraint might result in over-adjustment, potentially reducing statistical power by causing over-dispersion in a relatively homogeneous population.

In conclusion, this study identified 28 susceptibility loci for sarcoidosis among Europeans and American Africans. It revealed several critical pathways involved in sarcoidosis development, such as T cell and lymphocyte activation, leukocyte cell-cell adhesion, and activities related to the MHC protein complex. Additionally, obesity was found to be associated with an increased risk of sarcoidosis. The study also highlighted several potential therapeutic targets, including IL23R, TYK2, NFKB1, IL1RN, and IL2RB, which could pave the way for treatment strategies.

## Methods

### Ethics

The study complied with all relevant regulations governing the use of human participants and was conducted in accordance with the principles of the Declaration of Helsinki. Ethical permit is not necessary for the current study since this study is based on summary-level data. Participants in the FinnGen study provided informed consent for biobank research, with the study protocol (No. HUS/990/2017) approved by the Coordinating Ethics Committee of the Hospital District of Helsinki and Uusimaa (HUS). The UK Biobank received ethical approval from the North West Multi-centre Research Ethics Committee (approval number: 11/NW/0382), with all participants giving informed consent. The Million Veteran Program (MVP) was approved by the VA Central Institutional Review Board (IRB), and participants provided informed consent. Similarly, the All of Us Research Program was approved by the Institutional Review Board at Vanderbilt University Medical Center (VUMC), with participants also providing informed consent. Each study adheres to rigorous ethical guidelines to ensure the protection of participants and the integrity of the research.

### The FinnGen study

The FinnGen study is a large-scale ongoing genome-wide association initiative that harnesses the unique advantages of the Finnish population structure to explore genetic determinants of various human traits and diseases[45]. Drawing from the abundant phenotypic archives of national health databases and the extensive genotypic data amassed from biobank contributors, the FinnGen project aims to produce valuable insights into the biology underlying various health conditions and to promote the discovery of therapeutic targets. For this analysis, we utilized the FinnGen R11 GWAS summary-level data on sarcoidosis, which comprised 4,854 cases and 446,523 controls. Sarcoidosis cases were identified using the International Classification of Diseases (ICD) codes: 135 for ICD-8 and -9, and D86 for ICD-10. The GWAS associations were adjusted for age, sex, 10 principal components, and genotyping batch.

### The UK Biobank study

The UK Biobank is a large-scale biomedical database and research initiative, aiming to support diverse and groundbreaking scientific

discoveries to improve human health. The study has recruited ~500,000 UK individuals aged 40–69 years between 2006 and 2010 and collected extensive genetic and phenotypic information. Initially, participants lacking genetic data were excluded. Given few cases in non-European populations, we restricted our analysis to individuals of White-British descent, necessitating the exclusion of participants from other ethnic backgrounds. Sarcoidosis cases were identified using ICD codes 135 for ICD-8 and -9, and D86 for ICD-10. Our GWAS analysis comprised 795 sarcoidosis cases and 371,939 controls, with association tests adjusted for age, sex, the first 10 principal components, and genotyping batch.

### The Veterans Affairs Million Veteran program
The Million Veteran Program (MVP) was initiated in 2011 by recruiting participants aged 19 to over 100 years from 63 Veterans Affairs Medical Centers across the United States[46]. This study used data from 572,687 MVP participants (121,655 participants of African Americans and 451,032 participants of European ancestry). Sarcoidosis was defined by the Phecode Phe_697 that is curated groupings of ICD-9 and -10 codes with diagnostic data from electronic health records. A total of 1509 cases and 449,523 controls of European ancestry and 1827 cases and 119,828 controls of African Americans were included in the analysis. Associations were adjusted for age, sex, and top 10 population-specific genetic principal components.

### All of us research program
The All Of Us Research Program was started in 2018 with the aim of enrolling a diverse group of at least one million individuals across the USA to accelerate biomedical research and improve human health[47]. We acquired GWAS data from the Program's 'All by All' analysis conducted across participants with available short-read whole genome sequencing data. Similar to the MVP, sarcoidosis was also defined by Phecode Phe_697. A total of 396 cases and 99,021 controls of European ancestry and 374 cases and 40,033 controls of African Americans were included in the analysis.

### Genome-wide meta-analysis
Prior to meta-analysis, study-level quality control was conducted using GWASinspector[48] with default settings. This process included evaluating test statistic inflation, assessing skewness and kurtosis, checking for allele frequency mismatches, and performing allele harmonization. By filtering genetic variants with allele counts <50 in each study, we removed rare variants. We then employed the METAL[49] software to combine GWAS summary statistics data from the four studies under a fixed effects model to generate ancestry-specific and multi-ancestry meta-analysis summary statistics. The LDSC technique was implemented to evaluate the heritability of the trait, the genomic inflation factor (lambda), and the intercept[50]. Leveraging FUMA[51], we pinpointed independent genomic risk loci, conducting LD-based clumping against the backdrop of the 1000 Genomes Phase 3 dataset. The chosen significance threshold was $P < 5 \times 10^{-8}$, with a 500 kb window, and LD thresholds set at $r^2 = 0.6$ and $r_2^2 = 0.1$. The lead SNP presenting the locus is defined as the variant with the lowest $P$-value. The regional plot for GWAS association for each locus was used to visualize the genetic landscape and assess the surrounding genomic context. FUMA was further employed for the functional annotation of the earmarked loci. The MAGMA (Multivariate Analysis of Genomic Annotation) was used for gene mapping based on physical proximity and LD information. MAGMA tissue expression analysis was performed for gene expression with tissue-specific eQTL mapping with data from the Genotype-Tissue Expression (GTEx) v8 encompassing 53 tissue types[52]. For functional predictions, the tools incorporated the CADD score, which is a metric quantifying the deleteriousness of single nucleotide variants[53]. Variants with elevated CADD scores are generally considered more deleterious. We considered a locus to have a potentially

deleterious functional impact when it had a CADD score exceeding 25 (top 1% most deleterious substitutions in the human genome). Phenotypes associated with identified loci were collected by searching the GWAS Catalog (https://www.ebi.ac.uk/gwas/home).

### TWAS and pathway enrichment
The TWAS is a powerful post-GWAS analysis method for identifying significant gene-trait associations by modeling transcription-level regulation[54]. By leveraging expression quantitative trait loci data, TWAS excels in detecting functional genes influenced by disease-associated variants, offering valuable insights into disease mechanisms. We conducted a TWAS using GWAS summary statistics from European populations to minimize population structure bias by aligning with the reference data. To mitigate potential biases arising from variants with substantially different sample sizes, we restricted the analysis to genetic variants available in at least two studies. For this analysis, we employed the S-MultiXcan model[55] based on sarcoidosis GWAS summary-level data, which involved three key steps. First, we utilized the GTEx v8 expression dataset[56] as the reference during the training phase. The choice of GTEx v8 tissues was guided by the results of the MAGMA tissue expression analysis, which identified these tissues as most relevant for sarcoidosis. In this step, regulatory effect sizes of SNPs on gene expression were estimated using Elastic Net regression, a multivariate SNP - gene expression model. Elastic Net combines LASSO (L1) and Ridge (L2) regularization to handle multicollinearity among SNPs in linkage disequilibrium while selecting the most predictive SNPs within a 1 MB cis-window around each gene[56]. Next, predicted gene expression levels were computed using the sarcoidosis GWAS summary statistics as input. This step integrates SNP-gene weights from GTEx v8 with GWAS effect sizes to estimate tissue-specific gene expression levels. Finally, associations between predicted gene expression and sarcoidosis were evaluated using a multivariate statistical framework that accounts for cross-tissue correlations, as implemented in S-MultiXcan. Gene expression-sarcoidosis associations were considered significant at $P < 0.05$ after Bonferroni correction for multiple testing. Based on all significantly expressed signals obtained in TWAS, we performed pathway enrichment analysis to identify significantly enriched biological processes, cellular components, and molecular functions using the GO dataset.

### Mendelian randomization analysis
Because available GWASs on modifiable exposures mostly included individuals of European ancestry, we used sarcoidosis outcome data from our GWAS meta-analysis of 7554 individuals with sarcoidosis and 1,367,006 without sarcoid of European ancestry in the MR analyses. To evaluate the role of potential modifiable risk factors for sarcoidosis, we first conducted MR analyses to test the associations between sarcoidosis risk and 6 common modifiable risk factors including 2 measures of adiposity (BMI and WHR)[57] and 4 lifestyle factors (smoking initiation[58], alcohol intake[58], coffee consumption[59], and MVPA[60]).

Given that sarcoidosis is primarily considered an inflammatory disease, we conducted an inflammatory markers-wide MR analysis to evaluate the potential role of 66 circulating inflammatory markers in sarcoidosis. Genetic data on inflammatory markers were obtained from a GWAS meta-analysis of 6 studies including up to 59,969 participants of European ancestry. Detailed information can be found in a previously published paper.

In addition, we performed a hypothesis-free MR analysis to identify circulating blood proteins associated with sarcoidosis. GWASs of 4907 circulating proteins were available in the deCODE study including 35 559 Icelanders whose plasma proteins were measured using the SomaScan version 4 assay (SomaLogic). We further used an independent data source for circulating proteins from the Fenland study as the replication dataset. The study included 10,708 participants from the UK and measured blood proteins using the SomaScan version 4 assay

as well. Detailed information on the two studies can be found in the original papers[61–63]. These analyses aimed to provide evidence for therapeutic target selection for sarcoidosis given that blood proteins, as key regulators of molecular pathways, are frequently targeted for therapeutic interventions. Detailed information on GWASs of these exposures is presented in Supplementary Data 7.

Regarding genetic instrument selection, we selected single nucleotide polymorphisms (SNPs) associated with each exposure of interest at the genome-wide significance threshold ($P < 5 \times 10^{-8}$). For inflammatory markers and circulating proteins, only *cis*-acting SNPs (i.e., in or ±250 kb from the gene encoding the relevant protein) were selected to minimize directional pleiotropy. We then pruned SNPs by LD $r^2 < 0.01$ for modifiable risk factors and inflammatory markers to obtain independent genetic instruments. For circulating proteins, only the lead SNP that explains the most phenotypic variance was selected as the genetic instrument.

We harmonized data based on both effect and non-effect alleles. For exposures with one and two SNPs as instruments, we used the Wald-ratio and inverse variance weighted method under the fixed effects model to estimate their associations with risk of sarcoidosis, respectively. Otherwise, we used the multiplicative random effects inverse variance weighted method to obtain the MR association estimates. For exposures proxied by multiple SNPs, we conducted a series of sensitivity analyses, including the weighted median[64], MR-Egger[65], and MR-PRESSO[66] methods. We used Cochran's Q value to assess the heterogeneity among the SNP estimates. The MR-Egger intercept test served to identify directional pleiotropy. Multiple testing corrections were implemented using the Benjamini-Hochberg FDR method. For the identified associations of genetically predicted inflammatory markers and circulating proteins with sarcoidosis after FDR correction, we performed colocalization analysis to rule out bias caused by LD[67]. We regarded the posterior probabilities H4 from this analysis >70% as an indication of strong colocalization support. Given the possibility of more than one genetic hits (i.e., the SNP identified as having a high probability of being associated with the inflammatory protein and sarcoidosis) in one gene region, we further used SuSiE regression to detect colocalization evidence in a more accurate way[68]. Detailed information on the methods and model settings are displayed in Supplementary Methods. All analyses were based on two-sided tests and performed using the TwoSampleMR and MendelianRandomization packages in R (version 4.1.1).

### Reporting summary

Further information on research design is available in the Nature Portfolio Reporting Summary linked to this article.

## Data availability

This study used data from the All of Us Research Program's Controlled Tier Dataset, available to authorized users on the Researcher Workbench (https://www.researchallofus.org/). The summary statistics were acquired from their "All by All" large genetic association result database (https://support.researchallofus.org/hc/en-us/articles/27049847988884-Overview-of-the-All-by-All-tables-available-on-the-All-of-Us-Researcher-Workbench), the information on sarcoidosis GWAS in All of Us can be found in the table All by All phenotypes Google sheet [https://docs.google.com/spreadsheets/d/1sRxX4nzEPCKnzCgjJiwnt9h3HNrL_-zwpTjqZsikn34/edit?gid=1330947904#gid=1330947904] under the phecode 690. The UK Biobank data are available through application (https://www.ukbiobank.ac.uk/). Summary-level GWAS of sarcoidosis in the UK Biobank is available through the NHGRI-EBI GWAS catalog under accession number GCST90558258 [https://www.ebi.ac.uk/gwas/downloads/summary-statistics]. The summary-level data of FinnGen [https://storage.googleapis.com/finngen-public-data-r11/summary_stats/finngen_R11_D3_SARCOIDOSIS.gz] and MVP [https://ftp.ncbi.nlm.nih.gov/dbgap/studies/phs002453/analyses/GIA/] are publicly online. The GWAS and TWAS data

are deposited at OSF. The GWAS summary statistics are also available through the NHGRI-EBI GWAS catalog under accession codes GCST90503482, GCST90503483 and GCST90503484 [https://www.ebi.ac.uk/gwas/studies/GCST90503485]. Source data are provided with this paper.

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

## Acknowledgements

We want to acknowledge the participants and investigators of the UK Biobank study, the FinnGen study, the Veterans Affairs Million Veteran Program. This research was conducted using the UK Biobank study under application Number 72723. We gratefully acknowledge All of Us participants for their contributions, without whom this research would not have been possible. We also thank the National Institutes of Health's All of Us Research Program for making available the participant data examined in this study. S.Y. is supported by the American Heart Association Postdoctoral Fellowship (https://doi.org/10.58275/AHA.24POST1189614.pc.gr.190880). J.C. is supported by the "Co-PI" project from The Third Xiangya Hospital of Central South University (Number: 202401). J.Y. is supported by the NIHR Imperial Biomedical Research Centre (BRC). S.S.Z. is supported by a National Institute for Health and Care Research (NIHR) Clinical Lectureship and works in centres supported by Versus Arthritis (grant no. 21173, 21754 and 21755) and the NIHR Manchester Biomedical Research Centre (NIHR203308). E.V.A. is supported by the Swedish Research Council (2022-00826), the Swedish Heart-Lung Foundation (20200452), the Swedish Cancer Society (20 1299), the Sven and Ebba Hagberg Prize and Karolinska Institute. S.C.L. is supported by the Swedish Research Council (Vetenskapsrådet; 2019-00977), the Swedish Research Council for Health, Working Life and Welfare (Forte; 2018-00123) the Swedish Heart-Lung Foundation (Hjärt-Lungfonden; 20190247), and the Swedish Cancer Society (Cancerfonden).

## Author contributions

S.Y. and S.C.L. conceived and designed the study. S.Y., J.C., S.A., and S.M.D. contributed to data collection. S.Y. and J.G. undertook the statistical analyses. S.Y. created the data visualizations and authored the initial draft of the manuscript. S.Y., J.C., J.G., S.S.Z., J.Y., E.V.A., S.A., M.G.L., K.K.T., S.B., S.M.D., and S.C.L. contributed to data interpretation, offered significant intellectual insights to the manuscript, and approved its final version.

## Funding

## Competing interests

S.M.D. receives research support from RenalytixAI and in-kind research support from Novo Nordisk, both outside the scope of the current project. Other authors declare no conflict of interests.
