## [Transparent Peer Review file · Nature Communications]

Genome-wide association analysis reveals genetic loci, lifestyle factors, circulating biomarkers as risk factors for sarcoidosis

Corresponding Author: Dr Shuai Yuan

Version 0:

Reviewer comments:

Reviewer #1

(Remarks to the Author)

This report details a GWAS of sarcoidosis in the FinnGen study using controls from the UK biobank. Follow-up analyses include, among others, classification of the associated loci by function and potential deleterious effects. The authors also attempt to identify lifestyle risk factors using Mendelian randomization. Overall the study is well-defined and reasonably conducted. That said, the findings are largely incremental and given some of the details of the study population and the motivation for certain choices of study design, not of the significance one would expect in Nature Communications. More detailed comments are listed below:

- 1) The authors list the ICD codes used to identify patients in the FinnGen and UK biobank studies, but no details of the phenotype of these patients is given. The differing mechanisms underlying chronic/persistent versus resolving or even coincident sarcoidosis is well documented - particularly if also including Lfögren's. Therefore interpretation of the findings (except perhaps IL23) is difficult.
- 2) There are several genetic studies of sarcoidosis, including GWAS of various subsets and eQTL studies, even of the FinnGen cohort that are missing in the background.
- 3) Other biomarker and protein studies of sarc are also missing; some of which show association to biomarkers related to the genes identified, that are also not referenced.
- 4) The authors state adjusting for 10 principal components (presumably they mean the PCs derived when assessing ancestry informative markers). This seems very excessive given that the populations are reported to be relatively homogeneous. This could lead to over dispersion of the model. If however, the population is NOT homogenous and it requires 10 PCs to account for the variability then stratification may be required. Including more details about the population, the marker panel used for ancestry differentiation and a supplemental figure of the PCA analysis results are needed to justify whatever approach is used. Note, if these are all EU patients with intercontinental subdivision, a better strategy could be adjusting for global ancestry then for all significant peaks, include an adjustment for local ancestry as well.
- 5) There are several places in the manuscript that the authors reference a software package as opposed to an underlying statistical or computational method. This should be remedied.
- 6) There is no justification given for a 1:150 ratio of case to control. This too should be remedied. If it was just presumed that "more is better" than I suggest having a more specific reason, like matching to adjust for what might be heterogeneous ancestry - if that is the case.
- 7) This is somewhat minor, but several recent publications have illustrated the fallacy that BMI is a good measure of adiposity, particularly in certain populations. This needs to be at least mentioned.

Reviewer #3

(Remarks to the Author)

The manuscript by Yuan et al describes a GWAS meta-analysis of sarcoidosis in UK-Biobanks and FinnGen. They identify 14 genome wide loci of which 6 have not been previously reported.

Due to the relative rarity of the phenotype the investigators have been able to gather 5 194 cases in spite of two very large biobank studies. For GWAS studies this is not a very large sample, so the expected number of loci is relatively modest. Yet, the increase from eight to 14 is a significant improvement.

Technically I do not have much to comment, this is a very "routine" GWAS meta-analysis and follows the tradition in the field. My main concern is how the results is interpreted.

The one key benefit of biobank studies is that pleiotropy can be analyzed within the study sample. In the Methods they describe how they extract the information from the GWAs catalogue, yet I had a hard time to find the interpretation of these studies. A more surprising aspect is how the authors approached the question of pleiotropy. The FinnGen browser reports pleiotropy for every variant against thousands of disease endpoints, no additional analyses are needed. A large number of the previously identified, as well as the new loci are also contributing (either risk adding or protective) in many immune mediated diseases. Also some of the loci are associated with cardiovascular phenotypes like hypertension. Somehow the reader gets the impression that the text is written with a somewhat narrow sarcoidosis view. I strongly suggest that this aspect should be added throughout the manuscript and discuss the genetic findings in a wider perspective and the publicly available data in both UK Biobank and FinnGen should be used.

Version 1:

Reviewer comments:

Reviewer #1

(Remarks to the Author)

The increase in sample size is appreciated, but the results have not changed significantly from the previous version. The addition of the TWAS is an interesting attempt to make the findings more translational, but it is not clear what was actually done. The authors mention expression data on tissue, but was this data used as a reference panel from which to impute expression on some set of genotype data (the traditional definition of a TWAS), or was something else done. Clarity is needed to be able to assess the contribution of this to the work. New sections of text need to be checked for errors like "Figure 1a-c shows top genes with 8 genes...".

Reviewer #3

(Remarks to the Author)

The authors have now rerun the analyses and increased the case number significantly. This is a major improvement of the manuscript.

The way they present pleiotropy is not very easy to digest for the reader. Wouldn't a small summary table that lists the genome wide significant sarcoidosis loci and indicate the key pleiotropic traits and show which loci do not have pleiotropy be a more palatable way to present this. But this is a minor aspect.

I do not have further comments.

Version 2:

Reviewer comments:

Reviewer #1

(Remarks to the Author)

Thank you for the additional information about the TWAS. I have serious concerns about the applicability of the results using a general tissue panel as reference and a combined GWAS (across ancestries) and the data from which you imputed expression data. This has been shown to yield high levels of both false negative and false positive results. In general, it is hard to interpret the findings given that so many different diagnostic criteria, ancestries, and disease severity groups were combined.

Reviewer #4

(Remarks to the Author)

The authors conducted a meta-analysis of sarcoidosis using publicly available GWAS data from multiple biobanks, followed by a Transcriptome-Wide Association Study (TWAS) analysis. The GWAS analysis was well done and provides interesting results, my comments are thus primarily on the TWAS analysis.

Additional details for the TWAS analysis would be helpful. Specifically, it is unclear whether the authors are using pre-

computed weights from prior studies or computing their own predictor weights. If computing their own, the paper should specify how the expression and genotype data was processed and what "multivariate SNP ~ gene expression model" was actually employed (lasso? BLUP?).

It is also unclear how the GWAS data was used as input. Was the TWAS conducted using the GWAS summary data or the individual-level data? How did the analysis deal with variants that were only available or passed QC in one of the biobanks were treated -- variants with substantially different sample sizes should not be used in the TWAS analysis. How did the analysis deal with multiple populations in the target GWAS sample? TWAS/S-MultiXcan typically requires an LD reference panel that needs to match the target population, yet this study involved GWAS of multiple target populations so how was this addressed? A reasonable solution would be to only run the TWAS on European ancestry individuals using European ancestry LD reference panels; or to run the TWAS on each of the populations separately with population-specific reference LD data and then meta-analyze (or use a multi-ancestry TWAS fine-mapping approach like PMID 35931050).

The Methods state "the association between the predicted gene expression and sarcoidosis was assessed using various statistical association models" but provide no information on which statistical models were used.

It is likely the case that the authors handled these issues appropriately, but without these details the work is not currently reproducible.

Version 3:

Reviewer comments:

Reviewer #1

(Remarks to the Author)

While there are still concerns with the heterogeneity of the sample, the authors have addressed this in the manuscript and the additional details of the TWAS help alleviate those concerns as well.

Reviewer #4

(Remarks to the Author)

The authors have addressed my concerns.

To Editors and Reviewers:

We sincerely appreciate your thorough evaluation of our work, and the insightful feedback provided. During the revision process, we significantly enhanced the GWAS sample size by incorporating data from four large-scale studies: UK Biobank, FinnGen, MVP, and All of Us. This expanded dataset has increased the robustness and impact of our findings. We are confident that these revisions further align the manuscript with the high standards of **Nature Communications**, and we look forward to your feedback. We hope our revised submission will meet the criteria for publication.

Reviewer #1 (Remarks to the Author):

This report details a GWAS of sarcoidosis in the FinnGen study using controls from the UK biobank. Follow-up analyses include, among others, classification of the associated loci by function and potential deleterious effects. The authors also attempt to identify lifestyle risk factors using Mendelian randomization. Overall, the study is well-defined and reasonably conducted. That said, the findings are largely incremental and given some of the details of the study population and the motivation for certain choices of study design, not of the significance one would expect in Nature Communications. More detailed comments are listed below:

Response to the comment: We sincerely thank the Reviewer for reviewing our paper and providing constructive comments. Compared to previously published GWASs on sarcoidosis (1-3), our study increased the effective sample size by 2.6-fold and tripled the number of known loci. Our substantially larger and better powered analysis additionally facilitated downstream analyses, like TWAS and Mendelian randomization analysis to uncover novel etiological pathways and causal risk factors for sarcoidosis.

1) The authors list the ICD codes used to identify patients in the FinnGen and UK biobank studies, but no details of the phenotype of these patients is given. The differing mechanisms underlying chronic/persistent versus resolving or even coincident sarcoidosis is well documented - particularly if also including Lfögren's. Therefore, interpretation of the findings (except perhaps IL23) is difficult.

Response to comment 1: We thank the Reviewer for this insightful comment. We agree on the importance of distinguishing between sarcoidosis subtypes and their relevance in interpreting genetic findings. Unfortunately, the underlying mechanisms of different phenotypes are not well understood, which complicates the clinical classification of these patients. Moreover, our study lacked access to detailed subtype information for the patients included in these datasets, preventing us from conducting analyses specific to each subtype. We have acknowledged this limitation in the discussion section of the manuscript.

While the lack of subtype-specific data is a limitation, conducting a GWAS on overall sarcoidosis with a large sample size given the relative rarity of the disease can still yield valuable genetic insights particularly for shared signals, identify potential risk factors and therapeutic targets, and contribute to the development of personalized medicine. Our study also included data of two population groups, which may facilitate the exploration of genetic similarities and differences across populations. These findings can serve as a foundation for future research aimed at exploring the genetic basis of sarcoidosis in more detail.

Page 18: *“Third, some misclassification of the outcome was inevitable as sarcoidosis is difficult to diagnose and the diagnostic codes cannot be used to identify sarcoidosis phenotypes. Fourth, our study cannot differentiate genetic associations for sarcoidosis with varying clinical features (e.g., sarcoidosis with and without Löfgren's syndrome) due to data unavailability.”*

2) There are several genetic studies of sarcoidosis, including GWAS of various subsets and eQTL studies, even of the FinnGen cohort that are missing in the background.

Response to comment 2: We thank the Reviewer for raising this point. We have now described GWAS plus eQTL analysis on sarcoidosis in the Introduction part. For FinnGen, we have introduced the study in the Methods.

Page 3: *“In addition, genome-wide association study (GWAS) plus expression quantitative trait loci analysis identified more associated genes, like ADCY3 and CCL24.18,19”*

Page 4-5: *“The FinnGen study is a large-scale ongoing genome-wide association initiative that harnesses the unique advantages of the Finnish population structure to explore genetic determinants of various human traits and diseases.²⁴ Drawing from the abundant phenotypic archives of national health databases and the extensive genotypic data amassed from biobank contributors, the FinnGen project aims to produce valuable insights into the biology underlying various health conditions and to promote the discovery of therapeutic targets. For this analysis, we utilized the FinnGen R11 GWAS summary-level data on sarcoidosis, which comprised 4,854 cases and 446,523 controls. Sarcoidosis cases were identified using the International Classification of Diseases (ICD) codes: 135 for ICD-8 and -9, and D86 for ICD-10. The GWAS associations were adjusted for age, sex, 10 principal components, and genotyping batch.”*

3) Other biomarker and protein studies of sarc are also missing; some of which show association to biomarkers related to the genes identified, that are also not referenced.

Response to comment 3: Thank you for the comment. As suggested, we have discussed some previous studies on biomarkers for diagnosis and prognosis of sarcoidosis (1). In addition, we have provided more information on the data sources used in our MR analysis, particularly GWASs for inflammatory biomarkers. We further added a hypothesis-free protein-wide MR analysis to seek more circulating proteins associated with sarcoidosis.

Page 17:

“The relationship between serum biomarkers and sarcoidosis has been explored in many studies even though most studies were based on prevalent sarcoidosis.⁶⁴ A comprehensive review on biomarkers for the diagnosis and prognosis of sarcoidosis identified soluble interleukin-2 receptor (sIL-2R) and serum amyloid A (SAA) as the most sensitive markers for confirming the disease.⁶⁴ While our study focused on causal inference, our findings also support the involvement of the interleukin-2 pathway and serum amyloid A in sarcoidosis.”

Reference 1. Ramos-Casals M, Retamozo S, Sisó-Almirall A, Pérez-Alvarez R, Pallarés L, Brito-Zerón P. Clinically-useful serum biomarkers for diagnosis and prognosis of sarcoidosis. *Expert Rev Clin Immunol* 2019; 15: 391–405.

4) The authors state adjusting for 10 principal components (presumably they mean the PCs derived when assessing ancestry informative markers). This seems very excessive given that the populations are reported to be relatively homogeneous. This could lead to over dispersion of the model. If however, the population is NOT homogenous and it requires 10 PCs to account for the variability then stratification may be required. Including more details about the population, the marker panel used for ancestry differentiation and a supplemental figure of the PCA analysis results are needed to justify whatever approach is used. Note, if these are all EU patients with intercontinental subdivision, a better strategy could be adjusting for global ancestry then for all significant peaks, include an adjustment for local ancestry as well.

Response to comment 4: We thank the Reviewer for pointing this out. We acknowledge that this may seem excessive supposing the reported homogeneity of the population. However, due to limitations in our access to individual-level data, we are unable to change this setting or provide more detailed stratification based on global or local ancestry. Our intention was to ensure robust control for population stratification, which can sometimes necessitate the use of multiple PCs, even in seemingly homogeneous populations. We have added this as a limitation in the paper.

Page 18: *“Last, the fixed number of genetic principal components in the included studies could not be adjusted due to the unavailability of individual-level data. This constraint might result in over-adjustment, potentially reducing statistical power by causing over-dispersion in a relatively homogeneous population.”*

5) There are several places in the manuscript that the authors reference a software package as opposed to an underlying statistical or computational method. This should be remedied.

Response to comment 5: Thank you for this comment. We have checked this and updated the references.

6) There is no justification given for a 1:150 ratio of case to control. This too should be remedied. If it was just presumed that "more is better" than I suggest having a more specific reason, like matching to adjust for what might be heterogeneous ancestry - if that is the case.

Response to comment 6: We agree with the Reviewer. However, this ratio was unfortunately fixed since this GWAS meta-analysis was based on summary-level data. The imbalances in case-to-control ratios may impact the reliability and validity of GWAS results, majorly increasing type II error by lowering statistical power. However, this may not largely bias our findings since the employed statistical tools, like Regenie and SAIGE in these included GWASs incorporates several features that help mitigate these issues and provide reliable results.

7) This is somewhat minor, but several recent publications have illustrated the fallacy that BMI is a good measure of adiposity, particularly in certain populations. This needs to be at least mentioned.

Response to comment 7: Thank you for this point. We have now discussed this point in the paper.

Page 16: *“Using the MR approach, our research identified significant associations between genetically predicted BMI and sarcoidosis. A broader association between adiposity and sarcoidosis is further supported by suggestive links between genetically predicted waist-to-hip ratio (WHR) and sarcoidosis, although the effect size was smaller compared to that of BMI. This suggests that BMI may serve as a more robust proxy for obesity in distinguishing sarcoidosis risk at the population level.”*

Reviewer #3 (Remarks to the Author):

The manuscript by Yuan et al describes a GWAS meta-analysis of sarcoidosis in UK-Biobanks and FinnGen. They identify 14 genome wide loci of which 6 have not been previously reported. Due to the relative rarity of the phenotype the investigators have been able to gather 5194 cases in spite of two very large biobank studies. For GWAS studies this is not a very large sample, so the expected number of loci is relatively modest. Yet, the increase from eight to 14 is a significant improvement.

Response to the comment: We sincerely appreciate the positive feedback from the Reviewer. As suggested, we have increased the sample size by involving another two large-scale studies, the Million Veteran Program and All of US Research Program, making the number of cases to 9755. Our expanded GWAS meta-analysis uncovered 28 genome-wide significant loci associated with sarcoidosis.

Technically I do not have much to comment, this is a very “routine” GWAS meta-analysis and follows the tradition in the field. My main concern is how the results is interpreted. The one key benefit of biobank studies is that pleiotropy can be analyzed within the study sample. In the Methods they describe how they extract the information from the GWAS catalogue, yet I had a hard time to find the interpretation of these studies. A more surprising aspect is how the authors approached the question of pleiotropy. The FinnGen browser reports pleiotropy for

every variant against thousands of disease endpoints, no additional analyses are needed. A large number of the previously identified, as well as the new loci are also contributing (either risk adding or protective) in many immune mediated diseases. Also some of the loci are associated with cardiovascular phenotypes like hypertension. Somehow the reader gets the impression that the text is written with a somewhat narrow sarcoidosis view. I strongly suggest that this aspect should be added throughout the manuscript and discuss the genetic findings in a wider perspective and the publicly available data in both UK Biobank and FinnGen should be used.

Response to the comment: We thank the Reviewer for raising this point. We agree that FinnGen browser is a good tool to look at the phenotypes associated with identified loci. However, the GWAS Catalog may provide more information by involving data from different studies and populations. We therefore decided to continue to use this tool to collect information on traits associated with each identified locus. Instead of summing up number of traits associated with each locus, we have showed the exact phenotypes associated with the loci in the supplements. We have changed “pleiotropy” to “phenotypes associated with loci” to make this part more understandable. We have now updated the discussion part of the paper as suggested to reflect more wide aspect of sarcoidosis.

Page 7: *“Phenotypes associated with identified loci were collected by a search in the GWAS Catalog (<https://www.ebi.ac.uk/gwas/home>).”*

Page 11: *“Many of these loci were associated with other autoimmune diseases, such as Crohn's disease, rheumatoid arthritis, multiple sclerosis, and celiac disease as well as with immune cells, such as lymphocyte and eosinophil counts (Table S2).”*

Reviewer #1 (Remarks to the Author):

The increase in sample size is appreciated, but the results have not changed significantly from the previous version. The addition of the TWAS is an interesting attempt to make the findings more translational, but it is not clear what was actually done. The authors mention expression data on tissue, but was this data used as a reference panel from which to impute expression on some set of genotype data (the traditional definition of a TWAS), or was something else done. Clarity is needed to be able to assess the contribution of this to the work. New sections of text need to be checked for errors like "Figure 1a-c shows top genes with 8 genes...".

Response to the comment: We sincerely thank the Reviewer for evaluating our paper and providing valuable feedback. As the Reviewer noted, we conducted a traditional TWAS using reference data from GTEx v8 for spleen, whole blood, and lung tissues, where sarcoidosis-associated genes were enriched. We have now clarified the details of the TWAS analysis and revised the relevant text accordingly.

Page 7: *"Figure 1a-c shows top 20 gene-sarcoidosis associations from TWAS in spleen, whole blood, and lung. Among them, there were 8 genes (TCF19, CCDC88B, METTL21B, TSFM, UBASH3A, POR, UQCC2, and WHAMM) associated with sarcoidosis risk consistently across these tissues."*

Page 18: *"The TWAS is a powerful post-GWAS analysis method for identifying significant gene-trait associations by modeling transcription-level regulation.⁵⁷ By leveraging expression quantitative trait loci data, TWAS excels in detecting functional genes influenced by disease-associated variants, offering valuable insights into disease mechanisms. For this analysis, we employed the S-MultiXcan⁵⁸ model based on sarcoidosis GWAS summary-level data, which involves three key steps. First, we used the GTEx v8 expression dataset⁵⁹ as the reference during the training phase. The choice of GTEx v8 tissue data was guided by the results of MAGMA tissue expression analysis. In this step, we estimated the regulatory effect sizes of multiple SNPs on gene expression by fitting a multivariate $SNP \sim gene\ expression$ model using the GTEx v8 tissue data. Next, we predicted gene expression levels using sarcoidosis GWAS data as input. Finally, the association between the predicted gene expression and sarcoidosis was assessed using various statistical association models. The gene expression-sarcoidosis associations were deemed significant with P value <0.05 after Bonferroni correction."*

Reviewer #3 (Remarks to the Author):

The authors have now rerun the analyses and increased the case number significantly. This is a major improvement of the manuscript. The way they present pleiotropy is not very easy to digest for the reader. Wouldn't a small summary table that lists the genome wide significant sarcoidosis loci and indicate the key pleiotropic traits and show which loci do not have pleiotropy be a more palatable way to present this. But this is a minor aspect. I do not have further comments.

Response to the comment: We sincerely thank the Reviewer for the positive feedback and constructive points. We have now added a supplementary table to show the major pleiotropy (associations with autoimmune diseases) of identified loci as suggested.

Page 6: *“Associations with identified 28 sarcoidosis-associated loci in the GWAS Catalog database are shown in Table S2. Many of these loci were associated with other autoimmune diseases, such as Crohn's disease, rheumatoid arthritis, multiple sclerosis, and celiac disease (Table S3) as well as with immune cells, such as lymphocyte and eosinophil counts.”*

Supplementary table 3. Pleiotropic associations with autoimmune diseases for 28 sarcoidosis-associated loci in the GWAS Catalog database.

Locus	Lead SNP	Chr:Position	Nearest gene	Pleiotropy with autoimmune diseases
1	rs34017352	1:67132522	C1orf141, IL23R	Crohn's disease, acute anterior uveitis, ankylosing spondylitis, psoriasis
2	rs7549723	1:150569336	RN7SL600P	Not reported
3	rs72695390	1:167631783	RCS1	Not reported
4	rs2422255	1:172815258	RP1-15D23.2	Celiac disease
5	rs34977426	2:60731591	ATP1B3P1	Asthma, ulcerative colitis, celiac disease, Crohn's disease
6	rs546039326	2:96560781	ARID5A	Not reported
7	rs145955907	2:97725153	ZAP70	Not reported
8	rs10183338	2:110853239	ACOXL	Rheumatoid arthritis
9	rs67748055	2:198046702	PLCL1	Rheumatoid arthritis, asthma, eczema, Crohn's disease, systemic lupus erythematosus
10	rs10189685	2:202623726	AC009960.3	Not reported
11	rs10933330	2:230322452	SP140	Ankylosing spondylitis, Crohn's disease, psoriasis, primary sclerosing cholangitis, ulcerative colitis
12	rs17041517	3:4834831	ITPR1:AC018816.3	Not reported
13	rs1152002	3:12430372	PPARG	Not reported
14	rs7639471	3:112336864	CD200	Not reported

15	rs11567997	5:138289539	CDC25C	Not reported
16	rs4921493	5:159409099	AC008703.1	Ankylosing spondylitis, Crohn's disease, psoriasis, primary sclerosing cholangitis, ulcerative colitis
17	rs12195589	6:32476807	HLA-DRB9	Asthma, type 1 diabetes, autoimmune thyroid and liver diseases, multiple sclerosis, Graves' disease, psoriasis, celiac disease, overall autoimmune disease
18	rs873973	7:75837299	CCL24	Not reported
19	rs77112238	9:126425324	MVB12B	Not reported
20	rs77029323	10:62690119	ZNF365	Multiple sclerosis
21	rs11202051	10:80193615	ANXA11	Not reported
22	rs663743	11:64340263	CCDC88B	Rheumatoid arthritis, psoriasis vulgaris, multiple sclerosis, Crohn's disease, type 1 diabetes
23	rs7965287	12:57794913	TSFM	Multiple sclerosis
24	rs3184504	12:111446804	SH2B3	Type 1 diabetes, rheumatoid arthritis, autoimmune thyroid disease, overall autoimmune disease
25	rs11856316	15:82846540	HOMER2	Not reported
26	rs4788115	16:28986790	RP11-264B17.3:LAT	Not reported
27	rs34536443	19:10352442	TYK2	Rheumatoid arthritis, type 1 diabetes, psoriasis, systemic lupus erythematosus, inflammatory bowel disease, ankylosing spondylitis, and overall autoimmune disease
28	rs1893592	21:42434957	UBASH3A	Celiac disease, rheumatoid arthritis, type 1 diabetes

Chr, chromosome; SNP, single nucleotide polymorphism.

Reviewer #1:

Thank you for the additional information about the TWAS. I have serious concerns about the applicability of the results using a general tissue panel as reference and a combined GWAS (across ancestries) and the data from which you imputed expression data. This has been shown to yield high levels of both false negative and false positive results. In general, it is hard to interpret the findings given that so many different diagnostic criteria, ancestries, and disease severity groups were combined.

Response to the comment: We understand the Reviewer's concerns. However, we believe that incorporating multi-ancestry data adds significant value, aligning with the current trend in GWAS. Given the rarity of sarcoidosis, our study's sample size is critical to advancing understanding of the disease's genetic underpinnings. While we fully support further GWAS on sarcoidosis subtypes, we believe our paper contributes meaningfully to understanding both genetic and non-genetic risk factors for sarcoidosis. For TWAS, we have now revised it by conducting exclusively with the European GWAS, including only SNPs present in at least two studies. Additionally, we performed the same analysis using a GWAS meta-analysis combining Europeans and African Americans. Comparing the two gene sets revealed that ~90% of genes were consistently associated with sarcoidosis across both analyses. Pathway enrichment analysis conducted separately for each gene set showed that broadly similar pathways were enriched. Of note, we used multi-tissue model for TWAS, which took statistical power into consideration. We have now discussed the Reviewer's comments as limitations in the paper.

Page 13: *"Second, while our GWAS included both European and African American populations, our TWAS and MR analyses primarily focused on individuals of European ancestry. This decision was made to minimize population structure bias and address limitations in statistical power. Future studies are needed to extend these analyses to other populations for a more comprehensive understanding. Third, some misclassification of the outcome was inevitable as sarcoidosis is difficult to diagnose and the diagnostic codes cannot be used to identify sarcoidosis phenotypes. Fourth, our study cannot differentiate genetic associations for sarcoidosis with varying clinical features (e.g., sarcoidosis with and without Löfgren's syndrome) due to data unavailability. Although we fully support further GWAS on sarcoidosis subtypes, we believe our paper contributes meaningfully to understanding both genetic and non-genetic risk factors for sarcoidosis."*

Reviewer #4:

The authors conducted a meta-analysis of sarcoidosis using publicly available GWAS data from multiple biobanks, followed by a Transcriptome-Wide Association Study (TWAS) analysis. The GWAS analysis was well done and provides interesting results, my comments are thus primarily on the TWAS analysis.

Response to the comment: We thank the Reviewer for the positive feedback and have added more information on TWAS as suggested (described in detail below).

1. Additional details for the TWAS analysis would be helpful. Specifically, it is unclear whether the authors are using pre-computed weights from prior studies or computing their own predictor weights. If computing their own, the paper should specify how the expression and genotype data was processed and what "multivariate SNP ~ gene expression model" was actually employed (lasso? BLUP?).

Response to comment 1: Thank you for this insightful comment. We performed the transcriptome-wide association study (TWAS) using pre-computed gene expression weights from the GTEx v8 dataset, specifically for spleen, whole blood, and lung tissues where sarcoidosis-associated genes were enriched in MAGMA, as provided by the S-MultiXcan framework. These weights were derived using Elastic Net regression, a penalized linear regression model that integrates LASSO (L1 regularization) and Ridge regression (L2 regularization). We have now added this information in the paper as suggested.

Page 18: *“For this analysis, we employed the S-MultiXcan model (58) based on sarcoidosis GWAS summary-level data, which involved three key steps. First, we utilized the GTEx v8 expression dataset (59) as the reference during the training phase. The choice of GTEx v8 tissues was guided by the results of the MAGMA tissue expression analysis, which identified these tissues as most relevant for sarcoidosis. In this step, regulatory effect sizes of SNPs on gene expression were estimated using Elastic Net regression, a multivariate SNP ~ gene expression model. Elastic Net combines LASSO (L1) and Ridge (L2) regularization to handle multicollinearity among SNPs in linkage disequilibrium while selecting the most predictive SNPs within a 1-MB cis-window around each gene.⁵⁹ Next, predicted gene expression levels were computed using the sarcoidosis GWAS summary statistics as input. This step integrates SNP-gene weights from GTEx v8 with GWAS effect sizes to estimate tissue-specific gene expression levels. Finally, associations between predicted gene expression and sarcoidosis were evaluated using a multivariate statistical framework that accounts for cross-tissue correlations, as implemented in S-MultiXcan. Gene expression-sarcoidosis associations were considered significant at $P < 0.05$ after Bonferroni correction for multiple testing.”*

2. It is also unclear how the GWAS data was used as input. Was the TWAS conducted using the GWAS summary data or the individual-level data? How did the analysis deal with variants that were only available or passed QC in one of the biobanks were treated -- variants with substantially different sample sizes should not be used in the TWAS analysis. How did the analysis deal with multiple populations in the target GWAS sample? TWAS/S-MultiXcan typically requires an LD reference panel that needs to match the target population, yet this study involved GWAS of multiple target populations so how was this addressed? A reasonable solution would be to only run the TWAS on European ancestry individuals using European ancestry LD reference panels; or to run the TWAS on each of the populations separately with population-specific reference LD data and then meta-analyze (or use a multi-ancestry TWAS fine-mapping

approach like PMID 35931050).

Response to comment 2: Thank you for your insightful comment. Our TWAS was conducted using GWAS summary-level data. Since all significant signals identified in the GWAS of African Americans (which had a smaller number of cases and may have been underpowered) were also detected in the GWAS of Europeans, we re-ran the TWAS exclusively with the European GWAS, including only SNPs present in at least two studies. This analysis identified 97 genes associated with sarcoidosis. Additionally, we performed the same analysis using a GWAS meta-analysis combining Europeans and African Americans, filtering for SNPs present in at least two studies. This approach identified 75 genes associated with sarcoidosis. Comparing the two gene sets revealed that 67 genes (~90%) were consistently associated with sarcoidosis across both analyses. Pathway enrichment analysis conducted separately for each gene set showed that broadly similar pathways were enriched. We have now updated the corresponding text, results, and enrichment analysis in the manuscript to reflect these findings.

Page 7: *“Based on GTEx v8 gene expression data of these tissues, TWAS identified that genetically predicted expression of 60, 45, and 50 genes in spleen, whole blood, and lung respectively were associated with the risk of sarcoidosis after Bonferroni multiple testing correction (Table S4). Figure 1a-c shows the top 20 gene-sarcoidosis associations from TWAS in spleen, whole blood, and lung. Among them, 14 genes (ABT1, BTN2A2, C2orf74, CCDC88B, FAM213A, MDH2, METTL21B, PRSS16, RP3-473L9.4, TCF19, TSFM, UBASH3A, UQCC2, ZSCAN26) associated with sarcoidosis risk consistently across these tissues. Based on 97 unique TWAS signals, the GO (Gene Ontology) database enrichment analysis identified several relevant pathways (Figure 1d) potentially associated with sarcoidosis development, including regulation of T cell and lymphocyte activation, leukocyte cell-cell adhesion, and activities related to MHC protein complex (Figure 1e).”*

Page 18: *“We conducted a TWAS using GWAS summary statistics from European populations to minimize population structure bias by aligning with the reference data. To mitigate potential biases arising from variants with substantially different sample sizes, we restricted the analysis to genetic variants available in at least two studies.”*

3. The Methods state "the association between the predicted gene expression and sarcoidosis was assessed using various statistical association models" but provide no information on which statistical models were used.

Response to comment 3: TWAS uses a multivariable model to account for cross-tissue correlations. We have now removed “various” to avoid misunderstanding.

It is likely the case that the authors handled these issues appropriately, but without these details the work is not currently reproducible.

Response to the comment: We thank the Reviewer again for the positive feedback and have added more details of TWAS as suggested.

Reviewer #1 (Remarks to the Author):

While there are still concerns with the heterogeneity of the sample, the authors have addressed this in the manuscript and the additional details of the TWAS help alleviate those concerns as well.

Response to the comment: Thank you for the comment. We discussed heterogeneity from population and disease perspective in the paper.

Page 13: First, combining data from the different populations might introduce population stratification bias due to different genetic background. However, population principal components were taken into consideration before GWAS meta-analysis. In addition, this approach has been often used in previous studies 46,47 and limited heterogeneity was observed between studies.

Page 13-14: Fourth, our study cannot differentiate genetic associations for sarcoidosis with varying clinical features (e.g., sarcoidosis with and without Löfgren's syndrome) due to data unavailability.